# Effect of Work Environment on Presenteeism among Aging American Workers: The Moderated Mediating Effect of Sense of Control

**DOI:** 10.3390/ijerph17010245

**Published:** 2019-12-30

**Authors:** Tianan Yang, Hubin Shi, Yuangeng Guo, Xuan Jin, Yexin Liu, Yongchuang Gao, Jianwei Deng

**Affiliations:** 1School of Management and Economics, Beijing Institute of Technology, Beijing 100081, China; tianan.yang@bit.edu.cn (T.Y.); 18811363379@163.com (H.S.); guoyuangeng@foxmail.com (Y.G.); jinxuanonly@163.com (X.J.); a1057499672@163.com (Y.L.); gyc5896@163.com (Y.G.); 2Sustainable Development Research Institute for Economy and Society of Beijing, Beijing 100081, China; 3School of Management, Technical University of Munich, Uptown Munich Campus D, 80992 Munich, Germany

**Keywords:** sense of control, work environment, presenteeism, subjective social status, aging workers

## Abstract

With the rapid increase of aging workforces, companies worldwide are concerned with improving the health and working status of older workers. Sense of control is an important psychological variable in sociology but has attracted less attention in studies of occupational health and management. This study examined the association of sense of control with presenteeism among aging workers in the United States. Data from the Health and Retirement Survey were analyzed, specifically, 2308 observations in 2012. Structural equation modeling was used to assess work environment, sense of control (measured in relation to personal mastery and perceived constraints), and associations with presenteeism. The moderating effect of subjective social status on the association between sense of control and presenteeism was examined with a moderated mediation model. In the final structural equation modeling model, work environment was directly inversely associated with presenteeism, and work environment was significantly inversely associated with perceived constraints. There was a direct positive association between work environment and personal mastery, a direct positive association between perceived constraints and presenteeism, and a significant inverse association between personal mastery and presenteeism. The significant indirect effects between work environment and presenteeism were significantly mediated by sense of control. Subjective social status inversely moderated the relation between presenteeism and perceived mastery, a dimension of sense of control. To increase the performance of aging workers in the United States, managers should create a work environment that facilitates access to job resources, as this might improve personal sense of control, particularly among those with high subjective social status.

## 1. Introduction

### 1.1. Background

The global workforce crisis continues to worsen because of the overall labor shortage attributable to worldwide aging [1,2]. The United States Health and Retirement Survey (HRS) is a highly representative national survey that explores changes in labor force participation and health transitions individuals undergo near and after the end of their work lives. Since its launch in 1992, the study has collected information on income, work, assets, pension plans, health insurance, disability, physical health and function, cognitive function, and health care expenditures. Using unique and in-depth interviews, the HRS provides an invaluable and growing body of multidisciplinary data that researchers can use to address important questions about the challenges and opportunities of aging. The HRS was designed to examine relationships between health and economic factors during work transitions before and after retirement [3]. Although a number of important studies of worker aging and retirement used data from the HRS to examine the health of aging populations, the relationship between wealth and health, and the dynamic effects of health on labor force transitions [4,5], the impact of psychosocial factors was not considered [6,7]. To address this critical gap and better evaluate psychosocial characteristics, the HRS added the Participant Lifestyle Questionnaire (PLQ)—in the 2006 wave and after—to investigate these factors in aging populations [8].

With the increasing realization of the importance of psychosocial factors, measures to encourage aging workers to work at full productivity are now a global concern [9,10]. Sense of control is a key element to address these problems and is defined as beliefs individuals maintain regarding the extent to which they can shape the course of their own social outcomes. It comprises personal mastery and perceived constraints [11,12] and is frequently studied in sociology [13] but not in occupational, industrial, or organizational studies. Personal mastery refers to the extent to which an individual feels able to execute their goals, whereas perceived constraints are uncontrollable obstacles or factors that an individual sees as barriers to achieving their goals [14,15]. Workers with a greater sense of control address problems with confidence, thus alleviating associated pain and expediting issue resolution [16,17]. Once such workers decide on a task, they hold themselves responsible for their own success or failure and view misfortune as the result of personal mistakes. Evidence indicates that personal sense of control decreases slightly as people age [14,15,18]. Aging populations defined as age 60 years or older by the United Union [19] and age 65 years or older by the World Health Organization [20] are susceptible to loss of sense of control, and a consequent decrease in personal sense of control, which could reduce productivity. To provide empirical evidence on how to inspire the potential of aging workforces, this study investigated the association of sense of control with productivity loss and presenteeism at work among aging workers.

### 1.2. Literature Review and Hypotheses Development

Presenteeism, defined as potential productivity loss attributable to health problems or other events that adversely affect employees, is an important cause of productivity loss at workplaces [21,22,23,24,25,26,27]. Accumulating evidence has provided theoretical insights regarding the determinants of presenteeism for individuals [28,29]. Job stress and health have been thoroughly examined [27,30]; however, the study of sense of control has just begun. Sense of control was found to be empirically associated with higher job satisfaction and performance at work and with other positive outcomes [31,32]. Feeling able to control one’s behaviors and outcomes is critical in producing positive outcome, because events or situations are seen as manageable [33]. However, most empirical studies have focused on determinants of presenteeism at the individual level, rather than factors at the organization level [30,34,35] such as work environment, which might increase presenteeism by diminishing sense of control. As compared with employees working under conditions of discrimination or poor social support, those working in an environment with strong social support might be more willing to devote themselves to their organizations [21,36], particularly aging workers with a strong sense of control [37].

The effects of sense of control on presenteeism among aging workers differ in relation to social status [38,39]. Previous studies predicted that social class would be closely associated with reduced sense of control [40]. When people report that they are of lower subjective social status, they are indicating that they have fewer resources and are subordinate to others. As suggested in studies of power, status, and interdependence [41], self-perception of reduced resources and subordinate rank are associated with a diminished sense of personal control [42]. We contend that lower-status individuals are predisposed to feel little personal control. Because of this reduced sense of personal control, lower-status individuals will tend to downplay internal or dispositional causes of social behavior that imply control and influence [43]. As compared with traditional objective social status metrics (such as income and education), subjective social status (SSS)—an individual’s view of their position in the social hierarchy [44,45]—more comprehensively captures social status by providing an analogy to conceptualize a person’s place in society: those higher on a ladder are closer to the top; those lower on the ladder are closer to the bottom. Economically disadvantaged persons are unlikely to believe they are in control, because it is unrealistic and can lead to unwarranted self-blame for their circumstances [16]. However, aging workers with high subjective social status can effectively adjust to a situation and avoid presenteeism, even if they have a low sense of control at work. Therefore, we tested the hypothesis that subjective social status has a moderating effect on the association between sense of control and presenteeism, a relationship that might be explained by attribution theory.

Attribution theory classifies control perceptions as internal (life outcomes resulting from one’s own actions) or external (life outcomes resulting from outside forces) [46]. Therefore, it is necessary to analyze sense of control among aging workers through attribution theory, first proposed by Heider [47]. In social psychology, attribution is the process by which individuals explain the causes of behaviors and events. Models to explain this process are called attribution theory and focus on the causes of outcomes, or on means-ends relations. These models combine agent-ends and means-ends beliefs when assessing attributions about success and failure [48]. Such analysis entails assessment or manipulation of perceived causes and measurement of their effects on behavior, feelings, and expectations [49].

The work environment affects employees’ sense of control, which further influences work performance and induces presenteeism. In an attempt to control and adapt to an environment, people with varying levels of sense of control might consciously or unconsciously explain their social behaviors by external and internal attribution. External, or situational, attribution refers to an attempt to explain a person’s behavior in relation their situation. Internal, or dispositional, attribution is the process of ascribing behaviors to certain internal traits (such as abilities and motivations) rather than to external forces [50]. Given that the construct of sense of control is based on current perceptions, it is viewed as less of a personality characteristic and more of a contextually based feeling, which will likely change depending on one’s life circumstances [14]. People with a strong sense of control tend to make internal attributions and are more likely to associate their work situation with their own abilities, thus believing that they can control their reactions and that they are ready for the job [18].

We developed the following hypotheses and constructed a conceptual model (Figure 1).

**Hypothesis** **1** **(H1).**
*Work environment had a direct negative effect on presenteeism.*


**Hypothesis** **2** **(H2).**
*Sense of control (personal mastery and perceived constraint) mediated relationship between work environment and presenteeism.*


**Hypothesis** **3** **(H3).**
*Work environment can have a positive impact on personal mastery of employees and negative impact of their perceived constraints.*


**Hypothesis** **4** **(H4).**
*Subjective social status positively moderates the mediating effects of sense of control (personal mastery and perceived constraint).*


## 2. Materials and Methods

### 2.1. Sample

We analyzed cross-sectional data from the 2012 wave of the U.S. HRS, which measures psychosocial factors and productivity of the aging workforce. The HRS is widely recognized as the best source of publicly available data on the aging population of the United States. Funded by the National Institute of Aging and the Social Security Administration of the United States, the HRS was initiated in 1992. Persons older than 50 years were recruited by means of multistage sampling for participation in biennial surveys that assess the characteristics of the aging population. To avoid problems associated with aging and a decrease in the number of participants over time, new samples were recruited every 6 years [51,52]. In 2006, the HRS added a participant lifestyle questionnaire (PLQ) to their core biennial survey; the PLQ was administered to a random 50% of core panel participants. The PLQ was developed by the HRS Psychosocial Working Group and includes a perceived ability to work scale, sense of control scale (personal mastery and perceived constraints), work environment scale, and subjective social status scale [8].

In the 2012 wave of the HRS, PLQ data were available for 6932 participants, because only respondents in 2012 who had completed the face-to-face PLQ interview in 2006 were again rotated to this mode of data collection. Among these 6932 participants, 2308 (33.3%) were still employed and older than 50 years. Among those still employed, 2308 (100%) answered at least one question on the “PLQ 2012”, and the percentage of those with missing data was less than 9%. Presenteeism was ascertained only among employed persons, and data from these 2308 participants were analyzed in the present study.

### 2.2. Measures

Presenteeism was measured by using the perceived ability to work scale (PAWS). The PAWS is a reliable, effective tool for measuring perceived productivity loss and was reported to have acceptable psychometric characteristics in previous empirical studies and in US Health and Retirement Surveys [8,25,26,53,54]. It comprises four subjective items, such as “How many points would you give your current ability to work?”, and asks participants to rate their perceived ability on a scale of 0 to 10 (0 = simply unable to do the current job; 10 = current working ability is at its best; Cronbach alpha = 0.90). To ensure that scores reflect the magnitude of presenteeism, we changed the directionality of scores by subtracting the original PAWS scores from 10. Thus, higher values indicate greater presenteeism.

Work environment was measured with the five-item scale (1 = strongly disagree, 2 = disagree, 3 = agree, 4 = strongly agree, 5 = does not apply; Cronbach α = 0.75) from the social census conducted by the national anti-discrimination commission in 2002 [8]. Item 1 was reverse coded, and all responses of “5” were recoded as missing. Thus, higher values indicate a better work environment.

Sense of control was measured in two dimensions: personal mastery and perceived constraints [13]. Perceived constraints was measured with a five-item scale (six-point Likert scale: 1 = strongly disagree, 6 = strongly agree; Cronbach α = 0.86), and personal mastery was measured with a five-item scale (five-point Likert scale: 1 = strongly disagree, 5 = strongly agree; Cronbach α = 0.91). Higher values indicate greater personal mastery and perceived constraints.

Subjective social status was measured with the Cantril Ladder scale [55]. Respondents were asked to think of a ladder as representing their place in society by placing an X on a handrail ranging from one to seven. Those at the top are richest and have the best education and best jobs. Those high on the ladder are closer to the top; those low on the ladder are closer to the bottom.

### 2.3. Statistical Analysis

SPSS 25.0 (IBM Corp.: Armonk, NY, USA) and AMOS 21.0 (IBM Corp.: Armonk, NY, USA) were used for statistical analysis comprising descriptive analysis and path analysis. Structural equation modeling (SEM) was used to examine associations among work environment, sense of control, subjective social status, and presenteeism.

Before SEM, the expectation maximization method was used to impute missing values in the data before statistical analysis [56]. Correlation analysis was used to determine the significance of correlations between work environment, sense of control, subjective social status, and presenteeism.

Since structural equation modelling (SEM) has potential advantages of choice in analyzing path diagrams when these involve latent variables with multiple indicators [57]. To determine whether sense of control has an indirect effect on the relation between work environment and presenteeism in this study, SEM was used to test the mediating effect in our initial model using maximum likelihood method estimation [58]. The criteria used to evaluate the model were a root mean square error of approximation less than 0.08 and goodness-of-fit, normed fit, comparative fit, and Tucker–Lewis index values of 0.90 or higher [59]. These indicators have all been used to examine model fit in previous studies.

To test for the robustness of this model and to determine if standardized regression coefficients (β) differed by subgroup, we conducted subgroup analyses of two age groups and two gender groups. To ensure that the two subgroups were of equal size, age was categorized as old (>59 years; 59 was the median of the sample in this study) and young (≤59 years), based on the median (59 years) of the final sample. Gender was divided into two categories: male and female.

A nonparametric resampling procedure was used to assess mediation with SPSS INDIRECT Macros [60]. This bootstrapping technique is a powerful method for generating confidence intervals for indirect effects [61], which were defined as mediating when they were significant and the confidence interval did not include zero.

When significant mediation was established, the conditional indirect effect procedures recommended by Hayes and Preacher were used to determine if mediation depended on the level of the theoretically proposed moderator (i.e., subjective social status) [62,63]. One thousand bootstrapping resamples generated 95% confidence intervals, and the moderated mediation model was tested to determine if the conditional mediation model was significant for presenteeism [64].

## 3. Results

### 3.1. Preliminary Analysis

The demographic characteristics of the participants are shown in Table 1. There were 1004 men (43.5%) and 1304 women (56.5%); 135 (5.8%) participants were younger than 50 years, 1221 (52.9%) were age 51–60 years, 624 (27%) were age 61–70 years, 279 (12.1%) were age 71–80 years, and 49 (2.1%) were older than 80 years. Descriptive statistics for the measurement items for each latent variable, including means, standard deviations (SD), and missing values, are shown in Table 2.

Correlation coefficients (r) show the relation between latent variables (Table 3). Presenteeism was significantly negatively correlated with work environment (r = −0.27), perceived mastery (r = −0.27), and subjective social status (r = −0.24) and significantly positively correlated with perceived constraints (r = 0.31). There was a significant positive correlation between perceived mastery and working environment (r = 0.24) and a significant negative correlation between perceived constraints and working environment (r = −0.33). There was also a significant negative correlation between perceived constraints and perceived mastery (r = −0.38).

### 3.2. SEM Model

Before SEM, analysis of the measurement model showed that our model fit the data well: the goodness-of-fit index and comparative fit index values for all measurement models were between 0.916 and 0.989. In the final model, the fitness criteria indicated that the final model was appropriate (Figure 2). Work environment was directly inversely associated with presenteeism (β = −0.23, SE = 0.077; *p* < 0.001). There were a significant positive association between work environment and personal mastery (β = 0.32, SE = 0.058; *p* < 0.001) and a significant inverse association between work environment and perceived constraints (β = −0.38, SE = 0.055; *p* < 0.001). Moreover, we noted a direct positive association between perceived constraints and presenteeism (β = 0.19, SE = 0.034; *p* < 0.001) and a significant inverse association between personal mastery and presenteeism (β = –0.14, SE = 0.028; *p* < 0.001).

In SPSS, mediation analysis with model 4 of the PROCESS macro was used to determine whether personal mastery and perceived constraints mediated the association between work environment and presenteeism. A nonparametric resampling procedure was used to confirm mediation with model 4 of the PROCESS macro. This powerful, reasonable bootstrapping technique yielded confidence intervals for indirect effects. A significant indirect effect and mediation were considered present when the confidence interval did not include zero. Work environment was related to lower personal mastery, which in turn was related to lower presenteeism (point estimate for indirect effect = −0.0981, SE = 0.0200, 95% BCa CI = −0.1399 to −0.0618). Similarly, work environment was related to lower perceived constraints, which in turn was related to lower presenteeism (point estimate for indirect effect = −0.1738, SE = 0.0256, 95% BCa CI = −0.2249 to −0.1243).

Subgroup analyses (Table 4) showed that the model results are very robust in relation to subgroup. The path coefficients of subgroups were similar and significant.

### 3.3. Moderated Mediation Analysis

On the basis of significant results in mediation models and theoretical considerations, we used SPSS PROCESS Macros Model 14 to determine whether, how, and under what conditions a given effect occurs in relation to the moderating role of subjective social status. The overall test models are shown in Figure 1. Table 4 shows the results after subjective social status was entered into the model. The moderated mediation tested PROCESS Model 14 with presenteeism as the outcome variable. The product term of personal mastery and subjective social status had a significant predictive effect on presenteeism (β = 0.04, *p* < 0.05), but the product term of perceived constraints and subjective social status did not have a significant predictive effect on presenteeism (β = −0.01, *p* = 0.51). The specific indirect effects and standard errors of subjective social status, for different values, are shown in Table 4. Subjective social status significantly moderated the indirect effect of work environment on presenteeism. Specifically, the interactional effect between subjective social status and personal mastery decreased from lower to higher levels on the moderator. The effect on presenteeism was strongest when subjective social status was lowest (Table 5).

## 4. Discussion

This study utilized the sociological variable sense of control for an empirical study of the workplace. We investigated several relationships, including the influence of work environment on presenteeism, the mediating effect of the two dimensions of sense of control—personal mastery and perceived constraints—on the relationship between work environment and presenteeism, and the means by which subjective social status moderates the relationship between personal mastery and presenteeism. Work environment had a direct negative effect on presenteeism, and sense of control mediated the negative relationship between work environment and presenteeism. Personal mastery was negatively associated with presenteeism, and perceived constraints increased presenteeism. In addition, subjective social status had a significant negative role in the influence of personal mastery on presenteeism. However, subjective social status had no significant moderating effect on the influence of perceived constraints on presenteeism.

In accordance with our hypothesis, the two dimensions of sense of control—personal mastery and perceived constraints—mediated the relation between work environment and presenteeism. Previous studies reported that a strong sense of control benefited physical and mental health. Lachman and Firth found that people with strong beliefs regarding psychological control had lower incidences of acute and chronic diseases and better physical function and mental health [65]. Studies of enterprise management found that sense of control was important in reducing counterproductive behavior [37]. In addition, intervention measures that increase individual psychological resources may help employees maintain their work ability [66,67]. Attribution theory suggests that people with a strong sense of control tend to make internal attributions and are more willing to link their work status with their own abilities [50]. However, work environment affects employees’ sense of control [18] and thus their presenteeism. Therefore, we evaluated whether sense of control was a mediating variable between work environment and presenteeism, which was verified in this empirical study.

Research on sense of control in aging working populations is much needed. Lachman and Firth reported that belief in personal control of health and bodily functions declines with age [65]; thus, older adults feel less general control over their lives than do younger adults. The level of perceived constraints is significantly higher in older adults than in young and middle-aged adults [65]. The mean age of the American workforce is increasing, and more than 20% are now 55 years or older [51]. In addition, proposed reform of the American Social Security System will affect retirement benefits and increase the retirement age [68]. In light of trends in urbanization and aging of the population, policymakers in the United States and other countries should monitor and forecast sense of control among aging workers, to prevent strong negative work attitudes and early retirement. Moreover, enterprises should vigorously implement measures to improve sense of control and limit events that reduce it. When recruiting older employees, enterprise managers should carefully consider employee sense of control and should encourage older employees to be more active and to take the initiative in addressing problems at work. Older employees should be given the right to control their own work and should be given more autonomy in, their work. When sense of control is improved among aging workers, presenteeism improves, thereby augmenting enterprise productivity and performance.

Interestingly, when subjective social status improves, the inhibitory effect of personal mastery on presenteeism is significantly reduced. Sense of control is a cognitive foundation of mental health, and it greatly depends on the objective conditions of individuals in the social hierarchy [38]. Evidence indicates that subjective social status positively affects job satisfaction and, ultimately, work status [68]. Improvement in subjective social status changes sense of control and personal working status. Therefore, when we examined the effect of sense of control on presenteeism, we introduced the moderating variable subjective social status. Subjective social status was positively correlated with personal mastery and negatively correlated with perceived constraints, and had an inhibitory effect on presenteeism, which is consistent with the results of previous studies. Moreover, subjective social status significantly inversely regulated the relationship between personal mastery and presenteeism. As subjective social status increases in a population of aging workers, the inhibitory effect of personal mastery on presenteeism decreases. When a person’s social status is high but their sense of personal control is low, presenteeism is more likely. Senior managers have high subjective social status in a company, and their personal mastery has a much weaker inhibitory effect on presenteeism. We believe that enterprises should develop measures to enhance sense of control among aging workers, by encouraging them to take the initiative in solving problems at work, for example. In particular, they need to focus on management personnel, to prevent counterproductive behavior caused by marked aging-related declines in personal sense of control.

Our third finding was that work environment can enhance personal mastery of employees and limit their perceived constraints. Previous studies reported that personal characteristics, including socioeconomic status, were associated with sense of control [39]. However, studies of the antecedent variables of individual sense of control focused mainly on individual characteristics, and few considered variables at the organizational level. Humans naturally interact with their environment, to ensure that they experience the effects they expect and avoid unwanted effects [18]. Thus, organizational level studies are necessary. The likelihood of external and internal attribution changes in relation to the sense of control among aging workers. People with a strong sense of control tend to make internal attributions and are more likely to associate their work situation with their own abilities. They assume they can control their reactions and believe they are ready for the job. In contrast, aging workers with a weak sense of control are more inclined to external attribution, thus linking their own working conditions with the working environment and ascribing failure to a poor external working environment [69]. Therefore, we studied a new organizational factor in sense of control—work environment—and found that sense of control was affected by the work environment of aging workers. To make aging workers more confident and active in addressing problems, managers must adopt measures that reduce pressure on aging workers, promote fairness, and improve the work environment as part of enterprise management. Improvement of the work environment by managers might increase enthusiasm among aging employees, thereby augmenting enterprise productivity.

Finally, the work environment of aging workers can directly inhibit presenteeism. Previous studies of presenteeism focused on job requirements and resource models and suggested that support from supervisors and coworkers increases productivity [35,70]. Strong support from supervisors and coworkers is associated with higher productivity and less presenteeism [71], while work stress can trigger presenteeism [72]. Because aging workers seek to interact with the work environment, the work environment can directly inhibit their presenteeism. Improving the work environment can lessen counterproductive behaviors of aging workers. A new antecedent variable, presenteeism, was identified. This extends research on presenteeism and provides a potential direction for intervention in and reduction of presenteeism among aging workers.

Inevitably, this study has limitations. First, although our moderated mediating model is based on established theory, the cross-sectional design does not allow inferences regarding causality. Future studies should use a longitudinal study design or experiments. Second, some of the present data were perceptual rather than behavioral. Specifically, participants evaluated two variables—sense of control and work environment—on the basis of their subjective feelings rather than their own behavioral characteristics, which inevitably led to measurement deviation. There are differences between the perception of scales in eastern and western cultures [73,74], but due to the limitation of data acquisition, we did not test the perceptive scales in eastern cultures. Future studies should use objective methods to collect behavioral data on sense of control and actual observations of the work environment. Third, we focused on the aging work population of the United States, which limits the generalizability of the results. Nevertheless, the findings should prove useful in future research. Future studies should examine other countries, to ensure that the results are broadly applicable to those populations.

## 5. Conclusions

The importance of sense of control in the productivity of aging workers and their enterprises has been acknowledged in studies of occupational health and management. Improvements in the work environment of aging workers might enhance their sense of control by ensuring equal access to job resources, thereby significantly reducing productivity loss at workplaces.

## Figures and Tables

**Figure 1 ijerph-17-00245-f001:**
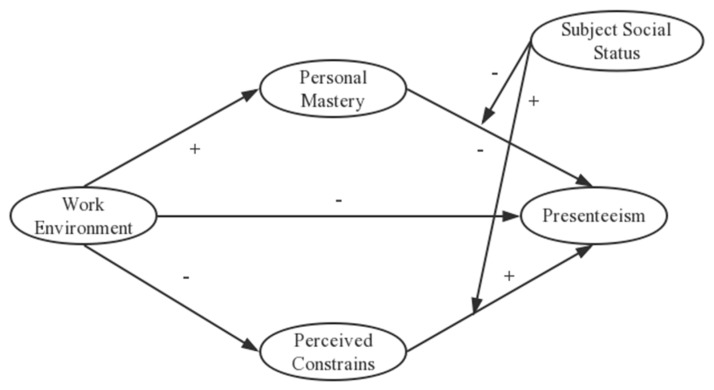
Proposed model of the relationship between work environment, presenteeism, and personal sense of control shows the effect of mediator resources, with subjective social status as moderator.

**Figure 2 ijerph-17-00245-f002:**
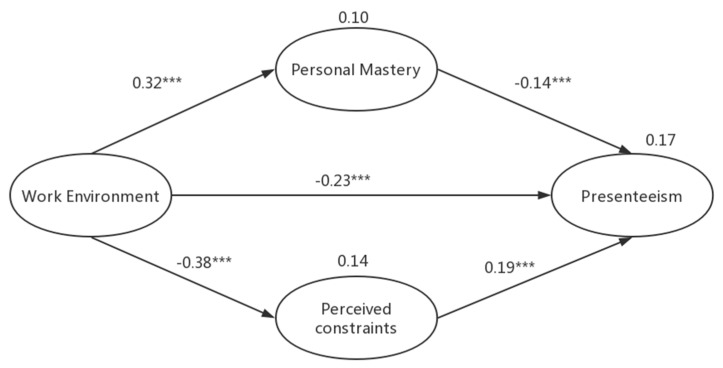
Mediator model of how work environment affects presenteeism (the numbers in the path are standardized regression coefficients and the numbers on the variables show variability, namely, root mean square error of approximation = 0.052, goodness-of-fit index = 0.955, comparative fit index = 0.961, Tucker–Lewis index = 0.954; *** *p* < 0.001).

**Table 1 ijerph-17-00245-t001:** Demographic characteristics of sample population (*n* = 2308).

	Characteristics	No.
Sex	Male	1004 (43.5%)
	Female	1304 (56.5%)
Age (years)	≤50	135 (5.8%)
	51–60	1221 (52.9%)
	61–70	624 (27.0%)
	71–80	279 (12.1%)
	>80	49 (2.1%)

**Table 2 ijerph-17-00245-t002:** Mean, SD, and percentage of missing values for each item.

Variables	Items	Mean	SD	Missing
Work Environment (WE)	(1) I have too much work to do everything well.	2.07	0.967	55 (2.4%)
	(2) I have a lot to say about what happens on my job.	2.88	1.015	64 (2.8%)
	(3) Promotions are handled fairly.	2.69	0.843	58 (2.5%)
	(4) I have the training opportunities I need to perform my job safely and competently.	3.15	0.746	50 (2.2%)
	(5) The people I work with can be relied on when I need help.	3.17	0.725	44 (1.9%)
Perceived Constraints (PC)	(1) I often feel helpless in dealing with the problems of life.	2.12	1.406	38 (1.6%)
	(2) Other people determine most of what I can and cannot do.	1.84	1.310	45 (1.9%)
	(3) What happens in my life is often beyond my control.	2.17	1.406	45 (1.9%)
	(4) I have little control over the things that happen to me.	1.94	1.302	38 (1.6%)
	(5) There is really no way I can solve the problems I have.	1.72	1.166	36 (1.6%)
Personal Mastery (PM)	(1) I can do just about anything I really set my mind to.	5.02	1.174	38 (1.6%)
	(2) When I really want to do something, I usually find a way to succeed at it.	5.09	1.133	39 (1.7%)
	(3) Whether or not I am able to get what I want is in my own hands.	4.76	1.267	49 (2.1%)
	(4) What happens to me in the future mostly depends on me.	4.93	1.259	40 (1.7%)
	(5) I can do the things that I want to do.	4.86	1.256	42 (1.8%)
Presenteeism (P)	(1) How many points would you give your current ability to work?	1.42	1.678	50 (2.2%)
	(2) Thinking about the physical demands of your job, how do you rate your current ability to meet those demands?	1.44	1.748	33 (1.4%)
	(3) Thinking about the mental demands of your job, how do you rate your current ability to meet those demands?	1.26	1.588	31 (1.3%)
	(4) Thinking about the interpersonal demands of your job, how do you rate your current ability to meet those demands?	1.40	1.657	39 (1.7%)
Subjective Social Status (SSS)	(1) Please mark an X on the rung on the ladder where you would place yourself.	6.37	1.59	200 (8.7%)

**Table 3 ijerph-17-00245-t003:** Mean, standard deviation, and correlations among study variables.

Variables (M, SD)	Items				
	WE	PC	PM	P	SSS
Cronbach α	0.75	0.86	0.91	0.90	
WE (2.98, 0.54)	1				
PC (1.96, 1.05)	−0.33 **	1			
PM (4.93, 1.04)	0.24 **	−0.38 **	1		
P (1.38, 1.46)	−0.27 **	0.31 **	−0.27 **	1	
SSS (6.37, 1.59)	0.31 **	−0.30 **	0.24 **	−0.24 **	1

** *p* < 0.01; WE, work environment; PC, perceived constraints on personal control; PM, perceived mastery; P, presenteeism; SSS, subjective social status.

**Table 4 ijerph-17-00245-t004:** Standardized regression weights (β) with *p*-values for the components of subgroup analyses.

Path	Sex				Age, Yeas			
	Male (n = 1004)		Female (n = 1304)		≤59 (n = 1242)		>59 (n = 1066)	
	β	*p*	β	*p*	β	*p*	β	*p*
WE to PC	−0.37	***	−0.39	***	−0.39	***	−0.37	***
WE to PM	0.24	***	0.37	***	0.37	***	0.35	***
PC to P	0.14	***	0.24	***	0.24	***	0.22	***
PM to P	−0.12	***	−0.16	***	−0.16	***	−0.17	***
WE to P	−0.24	***	−0.22	***	−0.22	***	−0.20	***

*** *p* < 0.001; WE, work environment; PC, perceived constraints on personal control; PM, perceived mastery; P, presenteeism.

**Table 5 ijerph-17-00245-t005:** Indirect effect of subjective social status (SSS) on presenteeism, by level of SSS.

Variable		BC 1000 BOOT			
SSS		P			
		IND	SE	LL95	UL95
Low	4.78	−0.12	0.03	−0.1818	−0.0588
Mean	6.37	−0.08	0.02	−0.1245	−0.0501
High	7.96	−0.05	0.03	−0.1032	−0.0025

Coefficients represent specific indirect effects and standard errors at different values of subjective social status, and the lower and upper bounds of 95% BC bootstrap confidence intervals for that effect, with 1000 bootstrap samples. Low signifies values at 1 SD below the mean, mean signifies values at the mean, and high signifies values at 1 SD above the mean. IND, indirect effects; P, presenteeism; SSS, subjective social status.

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
