# Peer review of "Effect of Work Environment on Presenteeism among Aging American Workers: The Moderated Mediating Effect of Sense of Control"

_ijerph, 2019, doi:10.3390/ijerph17010245_

Round 1
Reviewer 1 Report
My most sincere congratulations on a very interesting, well crafted and presented, and extremely important paper. With an aging society, investigation into issues related to an aging workforce are very badly needed. Your paper provides a springboard for that endeavor. I am hopeful that other authors will pick up the torch and follow suit in future investigations.
While I do not have any critical suggestions, I do have a question that you might consider responding to in the paper (or future works in this area). Did you consider the impacts of culture on the workforce's perceptions (i.e. Eastern as opposed to Western cultures)?
Author Response
Response: Thank you very much for your comment regarding the impacts of culture on workforce perceptions!
We believe that cultures in Asian and Western countries do differ [1,2], particularly in habits of thought, attention, and perception. However, because of limits in the data, we did not test this hypothesis. Future studies should use objective methods to collect behavioural data on sense of control and actual observations of the work environment. In addition, future research should continue to focus as much on cultural differences as on individual differences within cultures. We have revised the paper to reflect this (lines 410-417).
Sperber, A. D. Translation and validation of study instruments for cross-cultural research[J]. Gastroenterology, 2004, 126(supp-S1). Nisbett, R. E. Eastern and Western ways of perceiving the world[J]. Persons in context: Building a science of the individual. 2007,62-83.

Reviewer 2 Report
For the Proposed Model (Figure 1), between work environment, presenteeism, and personal sense of control, any reverse relationship between each mentioned variables? For the model (figure 1), for each variable i.e. WE, PC, PM, P, & SSS, any breakdown attributes from each of them? So that we can have a clear picture for the specific attribute in the relation, rather than a group as WE.
Author Response
Response: We have redrawn Figure 1 for the proposed model (Figure 1; please the attachment). We developed the following hypotheses.
Hypothesis 1. Work environment had a direct negative effect on presenteeism.
Hypothesis 2. Sense of control (personal mastery and perceived constraint) mediated relationship between work environment and presenteeism.
Hypothesis 3. Work environment can have a positive impact on personal mastery of employees and negative impact of their perceived constraints.
Hypothesis 4. Subjective social status positively moderates the mediating effects of sense of control (personal mastery and perceived constraint).
Sense of control is defined as beliefs individuals maintain regarding the extent to which they can shape the course of their own social outcomes. It comprises personal mastery and perceived constraints [1,2].
Work environment was measured with the five-item scale (1 = strongly disagree, 2 = disagree, 3 = agree, 4 = strongly agree, 5 = does not apply; Cronbach α = 0.75) from the social census conducted by the national anti-discrimination commission in 2002 [3]. Item 1 was reverse coded, and all responses of "5" were recoded as missing. Thus, higher values indicate a better work environment.
Further revisions were made in lines 56-60 and 130-141.
Figure 1. Proposed model of the relationship between work environment, presenteeism, and personal sense of control shows the effect of mediator resources, with subjective social status as moderator.
References
Infurna, F. J.; Mayer, A. The effects of constraints and mastery on mental and physical health: Conceptual and methodological considerations [J]. Psychology and Aging. 2015, 30, 432-448. Ward, M. Sense of Control and Self-Reported Health in a Population-Based Sample of Older Americans: Assessment of Potential Confounding by Affect, Personality, and Social Support[J]. International Journal of Behavioral Medicine. 2013, 20, 140-147. Smith, J. ; Fisher, G. ; Ryan, L. Psychosocial and Lifestyle Questionnaire 2006-2016 Documentation Report Core Section LB. Ann Arbor, MI, USA: The HRS Psychosocial Working Group; 2017.

Reviewer 3 Report
This paper investigates the association of sense of control with presenteeism among aging workers in the United States. The Health and Retirement Survey in 2012 is used for the analysis with an SEM approach. This paper seems to have some contributions to the literature but it has still large room for improvement.
1.Introduction/Literature Review
The authors did not separate the section of introduction and literature review, which provides some confusions related to the novelty of this paper. In particular, there has been a number of studies using the data set of the Health and Retirement Survey but this paper reviews a limited number of these studies. Inclusion of simple reviews for these papers help me understand the contribution of this paper clearly.
2.Hypothesis Development
There is no section for the development of empirical hypotheses as well. Developing more concrete Yes-No type hypotheses related to the role of sense of controls, especially based on extant literature may provide a better understanding for the contribution of the paper.
3. Model Description.
I am very confused about the empirical model used in the paper, particularly in terms of control variables. Furthermore, there is also limited motivation for the selection of SEM in the model estimation. Please fully specify the structural regression equations and the choice of control variables.
4. Robustness Checks.
This paper does not provide any robustness checks for the main results. Some robustness checks that use the gender of the survey participants and the age group of participants more detail (e.g. very old to old groups). Please provide some results related to robustness checks.
5. Minor comment: Abstract
I believe that the abstract unnecessarily includes beta coefficients and p-values. Please simplify the results.
Author Response
Responses to Reviewer 3
Introduction/Literature ReviewThe authors did not separate the section of introduction and literature review, which provides some confusions related to the novelty of this paper. In particular, there has been a number of studies using the data set of the Health and Retirement Survey but this paper reviews a limited number of these studies. Inclusion of simple reviews for these papers help me understand the contribution of this paper clearly.
Response: The introduction is now divided into two parts: (1) background and (2) literature review and hypothesis development.
Second, we reviewed a number of studies that used the dataset of the Health and Retirement Survey and, to augment our article [1-5], now refer to these papers.
Third, we added a literature review for each hypothesis in the introduction section, which should help improve understanding of this paper.
Revisions were made to lines 40-54. We hope these changes are satisfactory.
Hypothesis Development
There is no section for the development of empirical hypotheses as well. Developing more concrete Yes-No type hypotheses related to the role of sense of controls, especially based on extant literature may provide a better understanding for the contribution of the paper.
Response: After reviewing the extant literature, we added a literature review section for each hypothesis in the introduction section, which should improve understanding of our paper. For instance, (1) we introduce previous studies of sense of control and presenteeism [6,7,8], (2) we discuss previous studies showing that subjective social status had an effect on personal sense of control [9,10], and (3) we review attribution theory and construct a theoretical framework based on it [11,12,13,14].
Finally, on the basis of the above, we generated yes-no hypotheses, as follows.
Hypothesis 1. Work environment had a direct negative effect on presenteeism.
Hypothesis 2. Sense of control (personal mastery and perceived constraint) mediated the relationship between work environment and presenteeism.
Hypothesis 3. Work environment has a positive impact on personal mastery of employees and a negative impact on their perceived constraints.
Hypothesis 4. Subjective social status positively moderates the mediating effects of sense of control (personal mastery and perceived constraint).
Revisions were made to lines 76-85, lines 90-98, lines 109-117, and lines 130-137.
3.Model Description.
I am very confused about the empirical model used in the paper, particularly in terms of control variables. Furthermore, there is also limited motivation for the selection of SEM in the model estimation. Please fully specify the structural regression equations and the choice of control variables.
Response: Regression analysis is popular and effective in data analysis. However, in our study, we had to consider the path of included variables and latent variables that could not be directly measured. Structural Equation Modelling (SEM) has potential advantages over regression models, and SEM is thus "a priori the method of choice in analyzing path diagrams when these involve latent variables with multiple indicators” [15]. Revisions were made to lines 199-200.
Because of the large percentage of missing values for other demographic variables, we only used descriptive statistics for gender and age. We argue above that age affects sense of control [16] and thus only used age as a control variable.
Robustness Checks.
This paper does not provide any robustness checks for the main results. Some robustness checks that use the gender of the survey participants and the age group of participants more detail (e.g. very old to old groups). Please provide some results related to robustness checks.
Response: To test for the robustness of this model and determine if standardized regression coefficients (β) differed by subgroup, we conducted subgroup analyses of two age groups and two gender groups. To ensure that the two subgroups were of equal size, age was categorized as old (>59 years; 59 was the sample median) and young (≤59 years). Gender was classified as male and female.
Subgroup analyses showed that the model results were very robust in relation to subgroup. The path coefficients of subgroups were similar and significant. Revisions were made to lines 207-212 and 287-292.
Standardized regression weights (β) with p values for the components of subgroup analyses.
|
Path |
Sex |
Age, years |
|||||||
|
Male(N=1004) |
Female(N=1304) |
≤59(N=1242) |
>59(N=1066) |
|
|||||
|
β |
P |
β |
P |
β |
P |
β |
P |
||
|
WE to PC |
-0.37 |
*** |
-0.39 |
*** |
-0.39 |
*** |
-0.37 |
*** |
|
|
WE to PM |
0.24 |
*** |
0.37 |
*** |
0.37 |
*** |
0.35 |
*** |
|
|
PC to P |
0.14 |
*** |
0.24 |
*** |
0.24 |
*** |
0.22 |
*** |
|
|
PM to P |
-0.12 |
*** |
-0.16 |
*** |
-0.16 |
*** |
-0.17 |
*** |
|
|
WE to P |
-0.24 |
*** |
-0.22 |
*** |
-0.22 |
*** |
-0.20 |
*** |
|
*** p < 0.001; WE, work environment; PC, perceived constraints on personal control; PM, perceived mastery; P, presenteeism.
Minor comment: Abstract
I believe that the abstract unnecessarily includes beta coefficients and p-values. Please simplify the results.
Response: We have deleted the beta coefficients and p values and simplified the results. Revisions were made to lines 23-30.
References
Fisher, G. G.; Ryan, L. H.; Wang, M. Overview of the Health and Retirement Study and Introduction to the Special Issue[J]. Work, Aging and Retirement, 2018, 4(1):1-9. Smith; James, P. Racial and Ethnic Differences in Wealth in the Health and Retirement Study[J]. Journal of Human Resources, 1995, 30(4): S158-S183. Juster, T. F.; Suzman, R. An Overview of the Health and Retirement Study[J]. The Journal of Human Resources, 1995, 30(suppl). Willis, R. J.; Theory confronts data: How the HRS is shaped by the economics of aging and how the economics of aging will be shaped by the HRS[J]. Labour Economics, 1999, 6(2):119-145. University of Wisconsin-Madison. HRS REVIEW: PSYCHOSOCIAL VARIABLES. Retrieved from https://hrs.isr.umich.edu/sites/default/files/biblio/HRSReviewRyffPsychosocialVariables.pdf. Karanika-Murray, M.; Cooper, C.L. Presenteeism: An introduction to a prevailing global phenomenon. In L. Cooper & L. Lu (Eds.), Presenteeism at work. Cambridge: Cambridge University Press, 2018, 9-34. Lohaus, D.; Habermann, W. Presenteeism: A review and research directions[J]. Human Resource Management Review. 2018. Duffy, R. D.; Dik, B. J. Beyond the Self: External Influences in the Career Development Process[J]. Career Development Quarterly, 2009, 58(1):29-43. Lachman, Margie. E.; Locus of control in aging research: A case for multidimensional and domain-specific assessment. [J]. Psychology and Aging, 1986, 1(1):34-40. Sechrist, G. B.; Swim, J. K.; Stangor, C. When Do the Stigmatized Make Attributions to Discrimination Occurring to the Self and Others? The Roles of Self-Presentation and Need for Control. [J]. Journal of Personality and Social Psychology, 2004, 87(1):111-122 ; Paul, E. Development of the Work Locus of Control Scale[J]. Journal of Occupational Psychology, 1988, 61(4):335-340. Heider, F. The psychology of interpersonal relations[M]; The psychology of interpersonal behaviour. Penquin Books, 1978. Brewin, C. R.; Shapiro, D. A. Beyond locus of control: Attribution of responsibility for positive and negative outcomes[J]. British Journal of Psychology, 2011, 75(1):43-49. Kelley, H. H.; Michela, J. L. Attribution Theory and Research[J]. Annual Review of Psychology, 1980, 31(1):457-501. Gefen, D.; Rigdon, E.; Straub, D. Editor’s Comments: An Update and Extension to SEM Guidelines for Administrative and Social Science Research[J]. MIS Quarterly, 2011,35(2), iii. Mirowsky, J. Age and the Sense of Control[J]. Social Psychology Quarterly. 1995, 58, 31-43.Round 2
Reviewer 2 Report
For this version, it is fair to go.
Author Response
Dear reviewer,
Many thanks for your helpful suggestions on our manuscript!
Reviewer 3 Report
The authors revised the paper by accounting for my comments. One last thing I recommend is a more detailed illustration about the SEM, for instance, the exact equation formulation and the method of estimation.
Author Response
Response:
Thank you very much for your helpful comment on the SEM!
Model parameters can be estimated by several different methods, the usual methods include maximum likelihood (ML) method and generalized least square method [1]. ML is the default for many model-fitting programs. ML estimation is simultaneous, estimates are calculated all at once. If the estimates are assumed to be population values, they maximize the likelihood (probability) that the data (the observed covariances) were drawn from the population (the expected covariances). Maximum likelihood estimation methods are appropriate for nonnormally distributed data and small sample size. Thus, our model is estimated by the maximum likelihood method, which has been added in our manuscript (line 203).
Our model was proposed based on the following formulation:
x = ΛXξ+δ, (1)
y = ΛYη+ε, (2)
η = Βη+Гξ+ζ, (3)
x is the vector of exogenous observation variables; ξ is the vector of exogenous latent variables; ΛX is the relationship between exogenous observation variables, and exogenous latent variables, and the factor loading matrix of exogenous observation variable on the exogenous latent variable; δ is the error term vector of the exogenous variable; y is the endogenous observation variable vector; η is the endogenous latent variable vector; ΛY is the relationship between endogenous observation variable and endogenous latent variable and the factor loading matrix of endogenous observation variable on the endogenous latent variable; ε is the error term vector of endogenous variable. Both Β and Г are path coefficients; Β represents the relationship between endogenous latent variables; Г represents the impact of exogenous latent variables on endogenous latent variables; ζ represents the error term of structural equations.
Since this formulation might be quite complex to readers in our research field, we did not include the formulation of this model in our manuscript finally.
Hope our revision is satisfactory to you. Thank you so much again for your careful review for our manuscript!
Reference:
Schumacker, R.E., Lomax, R.G. A beginner's guide to structural equation modeling. LEA, 1996.
